# Essential Pieces of the Puzzle: The Roles of VEGF and Dopamine in Aging

**DOI:** 10.3390/cells14151178

**Published:** 2025-07-31

**Authors:** Melanie B. Thompson, Sanjay P. Tirupattur, Nandini Vishwakarma, Laxmansa C. Katwa

**Affiliations:** Department of Physiology, Brody School of Medicine at East Carolina University, Greenville, NC 27834, USA; thompsonmel24@students.ecu.edu (M.B.T.); tirupatturs24@students.ecu.edu (S.P.T.); vishwakarman18@students.ecu.edu (N.V.)

**Keywords:** aging, VEGF, dopamine, cardiovascular disease, neurodegenerative disease

## Abstract

Aging is a well-known, complex physiological process characterized by progressive functional decline and increased susceptibility to disease, particularly in the cardiovascular and nervous systems. While genetic and environmental factors can shape its advancement, molecular regulators such as vascular endothelial growth factor (VEGF) and dopamine signaling have emerged as critical factors in maintaining vascular and neural health. VEGF promotes angiogenesis and tissue repair, while dopamine, primarily recognized for its neuromodulatory roles, regulates vascular tone and appears to modulate VEGF activity. Despite substantial research on their roles in cardiovascular and neurodegenerative diseases, little is known about how VEGF and dopamine interact in the aging process, particularly in healthy versus unhealthy aging contexts. This review describes existing evidence on the independent and potentially complementary roles of VEGF and dopamine in aging, emphasizing their influence on maintaining or improving neurovascular health. It also explores how lifestyle interventions may be beneficial in modulating VEGF and dopamine signaling pathways in the aging population. By addressing the current knowledge gap surrounding VEGF–dopamine crosstalk, this review highlights the need for further investigation into their combined effects and targeting molecular interaction to unlock new research avenues for innovative strategies for healthy aging and the potential treatment of age-related diseases.

## 1. Introduction

Aging is a multifactorial biological process marked by the progressive loss of physiological integrity, accumulation of cellular and molecular damage, and increased risk of chronic disease, frailty, and mortality. Fundamental aging processes, such as genomic instability, telomere shortening, epigenetic alterations, proteostasis loss, and mitochondrial dysfunction, drive the onset of cellular senescence and chronic inflammation and impair tissue regeneration [1]. Environmental and behavioral factors, including physical activity and diet, further shape the severity and pace of aging, influencing both normal physiological decline and the inception of pathological conditions [2]. With increasing global life expectancy, research focus has shifted from simply extending lifespan to simultaneously improving life quality in elderly populations [1]. This shift calls for a deeper understanding of the physiological systems that regulate resilience and decline across multiple organ systems. The systems most affected by age-related dysregulation are the cardiovascular and nervous systems, which rely on tightly regulated molecular signaling to maintain homeostasis. Despite increasing recognition of the vulnerability of these systems with age, critical knowledge gaps remain in understanding the specific molecular interactions that influence their decline. In the molecular pathways involved in aging, vascular endothelial growth factor (VEGF) and dopamine have gained attention due to their critical roles in maintaining both cardiovascular and neurological function. VEGF, a central regulator of angiogenesis, ensures adequate tissue perfusion and repair by promoting endothelial cell growth and survival [3,4]. Dysregulation of VEGF signaling is implicated in various age-related conditions, including cardiovascular disease, macular degeneration, and impaired wound healing [5]. Additionally, beyond its well-known function in reward, mood, and motor pathways, dopamine contributes to vascular homeostasis by modulating blood flow and influencing endothelial responses [6].

Interactions between VEGF and dopamine signaling pathways become increasingly vulnerable with age, yet the interaction between VEGF and dopamine in this context remains insufficiently explored, particularly in the neuro- and cardiovascular systems. Although the referenced study focuses specifically on VEGF’s regulation of angiogenesis in the retina, it highlights the fundamental mechanisms of VEGF signaling under hypoxic and inflammatory conditions that may be relevant to systemic neurovascular aging [7]. Early evidence suggests that disruptions in either the VEGF or dopamine systems may influence the function of the other, potentially amplifying age-related deterioration in both the cognitive and cardiovascular domains. Given that cardiovascular disease remains the leading cause of death in older adults in the United States [8], clarifying the relationship between VEGF and dopamine may offer significant insight into disease progression and prevention. While addressing the individual roles they play in human physiological function, this review will examine VEGF and dopamine as interconnected signaling pathways with broader implications for neuro- and cardiovascular health. By stressing their shared roles in maintaining physiological resilience, this review aims to emphasize the importance of exploring VEGF–dopamine interaction in normal and pathological aging contexts. The subsequent section provides a detailed overview of the individual and overlapping physiological functions of VEGF and dopamine to help clarify how they may contribute to the progression and modulation of age-related changes.

## 2. VEGF and Dopamine

VEGF is a central signaling molecule in vascular biology, best known for its role in promoting angiogenesis through binding to the endothelial receptors VEGFR-1 and VEGFR-2 [9]. Under normal physiological conditions, VEGF supports wound healing, hypoxia adaptation, and organ development by enabling endothelial proliferation, migration, and survival. It is especially critical during embryogenesis and early postnatal growth, guiding vascular formation and tissue patterning. However, VEGF signaling and expression decline in aging tissues, resulting in reduced angiogenesis, diminished capillary density, and impaired tissue perfusion, particularly in organs with high metabolic demands such as the brain and heart [10]. Building on this understanding of VEGF’s role in vascular and neural maintenance, additional studies have explored its effects on specific models of neurodegeneration.

In a study modeling Parkinsonian neurodegeneration, VEGF administration in a 6-hydroxydopamine (6-OHDA)-induced rat model resulted in significant neuroprotection of dopaminergic neurons and improved motor function. Yasuhara et al. demonstrated that VEGF treatment enhanced the survival of tyrosine-hydroxylase-positive neurons and activated intracellular signaling pathways such as PI3K/Akt and MAPK/ERK, supporting cellular resilience and metabolic stability [11]. These findings suggest that VEGF can directly protect dopamine-producing neurons from degeneration by promoting angiogenesis, increasing nutrient supply, and activating intracellular survival pathways. Such neurovascular support may be critical for maintaining dopaminergic function in the aging brain and offers a strong rationale for further research into VEGF-based neuroprotection strategies. Although the study was conducted in a rodent model, the conserved nature of VEGF signaling pathways and dopaminergic neurobiology across mammalian species supports the relevance of these findings to human aging and neurodegenerative conditions, particularly in Parkinson’s disease [11].

Dopamine, a catecholamine neurotransmitter synthesized in the substantia nigra and ventral tegmental area of the brain, acts through D1–D5 receptors to regulate motor control, cognition, and reward processing. With aging, dopamine synthesis, receptor density, and signaling efficiency decline, which contribute to cognitive slowing, reduced motor coordination, and increased vulnerability to neurodegenerative diseases such as Parkinson’s and Alzheimer’s [12,13,14]. In addition to neurological functions, dopamine plays a key role in vascular regulation. It can modulate peripheral blood flow and inhibit VEGF-driven angiogenesis through D2-like receptor (D2R, D3R, D4R)-mediated suppression of endothelial proliferation and VEGFR phosphorylation [6]. Although this regulatory effect may have therapeutic value in limiting tumor angiogenesis, the age-related decline in dopaminergic tone could impair vascular function by removing an essential inhibitory function. Notably, while the cited study primarily addresses VEGF’s protective role in dopaminergic neuron survival, additional research suggests that dopamine may also exert tumor-suppressive effects by inhibiting angiogenesis and modulating tumor microenvironments [15]. However, this remains an emerging area of investigation. Conversely, VEGF has been shown to promote the survival of dopaminergic neurons [10], suggesting a reciprocal relationship with therapeutic relevance in neurodegenerative conditions. This mutual regulation positions VEGF and dopamine as co-dependent factors essential to neurovascular health throughout the lifespan of an individual.

By highlighting the interdependent roles of VEGF and dopamine, this article underscores a pressing need to further investigate their combined impact on the aging process. Their influence spans childhood to adulthood, from supporting basic tissue development to potentially mitigating age-related dysfunction. Understanding how these pathways contribute to normal and pathological aging points toward promising targets for intervention, making this exploration both timely and necessary for advancing aging research.

## 3. Interaction Between VEGF and Dopamine

The crosstalk between VEGF and dopamine represents a critical but underexplored area in aging biology, particularly within the neurovascular system. VEGF independently regulates angiogenesis and neural support, while dopamine governs neurotransmission and vascular tone. Their convergence may shape how vascular and neurological decline progresses over time. Additionally, these two signaling molecules operate at the interface of the circulatory and nervous systems, and their mutual regulation is essential for maintaining neurovascular integrity across an individual’s lifespan. This interaction supports normal physiological processes, including synaptic plasticity, cerebral perfusion, and endothelial stability. However, as both VEGF and dopamine undergo functional decline with age, disruptions in this balance may contribute to the emergence and acceleration of pathological aging. To visualize this relationship, Figure 1 depicts the general trajectory of VEGF and dopamine activity across the human lifespan. Both molecules present developmental increases during childhood, stabilization in early to mid-adulthood, and decline during advanced age [16,17,18,19,20,21]. This decline aligns with the onset of neurodegeneration and elevated cardiovascular risk, as marked by inflection points in Figure 1 [22,23,24,25]. The parallel reduction in VEGF and dopamine levels suggests an intertwined physiological relationship that supports shared pathological mechanisms in age-associated diseases.

As previously mentioned, dopamine has been shown to inhibit VEGF-induced angiogenesis via D2 receptor signaling, which suppresses endothelial cell proliferation and reduces vascular permeability [6]. Although this regulatory effect is beneficial in contexts like tumor suppression, where limiting abnormal vessel growth is desirable, the age-related loss of dopaminergic tone may remove this regulatory balance. The resulting disruption may lead to excessive or misdirected angiogenesis in aging organs, particularly in the brain and heart. Conversely, VEGF plays a neuroprotective role by supporting the survival and function of dopaminergic neurons. Experimental models have shown that exogenous VEGF administration enhances the viability of these neurons in degenerative conditions. For example, in a study modeling middle cerebral artery occlusion (MCAO), VEGF was administered to rats via intraventricular infusion and compared with rats that received artificial cerebrospinal fluid (aCSF) [26]. Metrics such as mean arterial blood pressure, arterial partial pressure of O_2_ or CO_2_, arterial blood pH, and blood glucose concentration were tested at baseline and 30 min following MCAO induction. Sun et al. demonstrated that VEGF treatment was linked to a reduction in hallmarks of cellular injury, including cell shrinkage and DNA strand breaks. Furthermore, over a 4-week period, the infarct volume in the group of VEGF-treated rats was 35% lower than that of the aCSF group, while the VEGF-treated rats simultaneously exhibited higher neurological function [26]. Such findings emphasize the complementary relationship between VEGF and dopamine in preserving neurovascular function.

During healthy aging, regulating these pathways supports adaptability, tissue repair, and vascular tone. In contrast, dysregulation may intensify vascular rigidity and cognitive deterioration and reduce regenerative capacity. Despite ample evidence on their roles, relatively few studies have explored the synergistic impact of VEGF and dopamine in aging-related decline [27,28]. This gap in literature limits understanding of how the combined dysfunction of VEGF and dopamine influences the progression of neurovascular disorders. Recognizing this interdependency offers a new lens to interpret the biology of aging and underscores the value of targeting this area in future therapeutic strategies. The following section will apply this conceptual framework to specific pathological outcomes in the cardiovascular and neurological systems.

## 4. Pathological Aging and Disease Context

Pathological aging is marked by the onset of chronic diseases that arise when core systems regulating vascular integrity and neural function begin to fail. The cardiovascular and nervous systems are particularly vulnerable due to their dependence on angiogenic and dopaminergic regulation. While prior sections have outlined the physiological roles of VEGF and dopamine, this section emphasizes how their dysregulation jointly contributes to age-related diseases. In the cardiovascular system, diminished VEGF signaling limits angiogenesis and capillary maintenance, increasing the risk of ischemic events such as myocardial infarction and stroke [10]. Simultaneously, reductions in dopamine disrupt vascular tone and impair endothelial function, exacerbating hypertension and limiting recovery following vascular injury [6]. In the nervous system, a deficiency in VEGF compromises neuroprotection and tissue oxygenation, particularly in regions dependent on dopaminergic signaling. Dopamine plays a role in mood and motor function and mediates neurovascular coupling, the process that ensures local blood flow matches neural activity. With aging, the disruption of this mechanism contributes to perfusion mismatch, increased oxidative stress, and a breakdown in blood-brain barrier integrity [4]. This interconnected decline within the nervous system mirrors similar vulnerabilities in the cardiovascular system, suggesting that both networks are affected by reductions in VEGF and dopamine signaling. Notably, VEGF’s influence on synaptic plasticity and neurogenesis in the adult brain underscores its relevance to maintaining cognitive flexibility and memory consolidation into later life. Recent research has expanded the understanding of VEGF’s role beyond angiogenesis, highlighting its direct influence on neuronal function. The article by Aksan and Mauceri provides compelling evidence that VEGF family members, particularly VEGF-A, play critical roles in neuronal development, morphology, and survival [29]. VEGF is shown to regulate neurite extension, dendritic complexity, and synaptic remodeling, processes essential for healthy cognitive and neurological function [29]. These insights position VEGF not merely as a vascular growth factor but as a promising and multifaceted therapeutic target with the potential to slow neurodegenerative processes and cognitive decline. Exploring how VEGF’s dysregulation and declining dopamine levels may impact broader neurovascular dysfunction across aging populations becomes increasingly important.

### 4.1. VEGF and Dopamine in Parkinson’s and Alzheimer’s Disease

Emerging research continues to build on this foundation, offering more targeted insights into mitochondrial dysfunction, dopaminergic vulnerability, and the potential reparative roles of VEGF. A study by Caballero et al. provides evidence that VEGF contributes to the survival of dopaminergic neurons by promoting mitochondrial function, reducing oxidative stress, and enhancing cellular resilience in Parkinson’s disease models [30]. Although the interaction between VEGF and dopamine is not extensively covered, the study supports the idea that VEGF can mitigate some of the mitochondrial and oxidative-stress-related damage linked to dopaminergic degeneration in Parkinsonian pathology. In Alzheimer’s disease, decreased VEGF levels have been linked to reduced cerebral angiogenesis and neurogenesis, contributing to impaired vascular support for neural tissue. At the same time, dopamine dysfunction in Alzheimer’s has been shown to affect synaptic activity, learning, and memory consolidation, compounding the neurodegenerative process. For example, a study by Wang et al. demonstrates that VEGF-induced angiogenesis ameliorated memory impairment in an APP transgenic mouse model of Alzheimer’s disease [31]. While the study was conducted in a mouse model, the observed mechanisms enhanced cerebral perfusion and synaptic preservation, while reducing amyloid pathology. These parallels strongly reinforce VEGF’s potential as a therapeutic agent in combating Alzheimer’s-related cognitive decline.

### 4.2. VEGF and Dopamine in Cardiovascular Diseases

There is limited research on the intersection of VEGF and dopamine in the context of cardiac function and cardiovascular disease. However, here we have outlined the studies that have so far focused on the separate roles of VEGF and dopamine in cardiac fibrosis. In parallel to these neural findings, a study by Katwa et al. highlights the role of cardiac myofibroblasts as a novel source of VEGF and its receptors Flt-1 and KDR, also known as VEGFR2, emphasizing VEGF’s contribution to angiogenesis and tissue remodeling in the cardiovascular system [32]. Furthermore, myofibroblasts are particularly important in cardiovascular function because they are central to wound healing, extracellular matrix remodeling, and scar formation after myocardial injury. By secreting VEGF, myofibroblasts help orchestrate the repair process by promoting new vessel growth, aiding in the revascularization of ischemic tissue, and supporting structural stability in damaged myocardium. The study demonstrated that VEGF expression by these fibroblasts may help mediate vascular repair and myocardial adaptation under stress, thereby playing a protective role in cardiovascular aging [20]. The article further emphasizes that the mutual decline of these molecules not only worsens tissue perfusion but also disrupts the balance and metabolic support essential for cognitive health.

Unlike VEGF, the discussion of intracardiac dopamine receptor expression is a relatively new topic, but the emerging role of dopamine receptors in fibroblasts and other heart cell types is of interest in the context of aging. In a notable study performed in wild-type (WT) and dopamine receptor 3 knock-out (D3KO) mice, Kisling et al. not only demonstrated dopamine receptor 1 (D1R) and dopamine receptor 3 (D3R) expression in mice hearts, but also that D3R dysfunction reduces cardiac fibroblast proliferation and migration and consequently inhibits a key element in the wound healing process of the heart [33]. Additionally, Byrne et al. review several studies on dopamine receptor 1 (D1R) and dopamine receptor 3 (D3R) in the heart, and the studies conveying the effect of dysfunctional intracardiac dopamine signaling are of particular interest [34]. Regulation of blood pressure and development of fibrotic tissue were comparable in young and old mice with the loss of D3R function in rodent models, which shows that functional dopamine is important in maintaining healthy cardiac function [35].

### 4.3. VEGF and Dopamine in Males vs. Females

While the involvement of both VEGF- and dopamine-mediated pathways in the progression of neurodegenerative and cardiovascular disease is widely implicated, emerging evidence also suggests that these signaling pathways are not uniform across the sexes. Biological sex influences key aspects of aging and neurovascular disease development. A review describes years of evidence showing that dopaminergic signaling differs significantly between males and females, both in healthy individuals and those with neuropsychiatric illness. The authors emphasize that sex-based heterogeneity in dopamine receptor biology, including D1-like and D2-like receptors and their heteromeric complexes, can influence disease vulnerability and therapeutic efficacy [36]. These differences also exist in VEGF signaling, particularly with vascular function and in cardiovascular diseases. In a study using human pluripotent stem cells (hPSCs), researchers observed sex-based disparities in endothelial progenitor differentiation efficiency. Male hPSCs responded well to a GSK3 inhibitor-based protocol, but female cells required supplemental VEGF treatment to achieve comparable differentiation. This disparity is likely due to differential endogenous VEGF expression, underscoring how sex-specific molecular environments can influence responsiveness to angiogenic cues [37]. Furthermore, using a mouse model exposed to different durations of weight gain and loss, another study found that male and female mice show distinct ligand and receptor profiles even with identical dietary intervention and pronounced sex-specific angiogenic responses [38]. These findings show the importance of incorporating sex-specific angiogenic cues into models of VEGF signaling, especially for aging-associated metabolic diseases involving vascular remodeling.

Estrogen and other sex hormones modulate VEGF pathways, contributing to sex-specific patterns in diseases like coronary artery diseases and microvascular ischemia. Moreover, conditions such as hot flashes and night sweats are linked to menopause and reflect altered dopamine-mediated vascular reactivity [39]. A recent review by Collignon et al. highlights how sex hormones influence cerebrovascular development as well, noting key pathways like VEGF and Wnt signaling. Specifically, the paper connects these vascular processes to neurodevelopmental disorders with strong sex biases. Conditions such as autism spectrum disorder and ADHD are associated with altered prenatal testosterone and estrogen levels, which may not only affect VEGF-mediated vascular development but also likely influence dopaminergic signaling given the shared reliance on proper neurovascular support [40]. This, furthermore, strengthens the intersection of VEGF and dopamine in disease development and stresses the importance of further exploration. Considering these sex differences in VEGF and dopamine signaling is critical for developing more precise diagnostic tools and therapeutic interventions, especially since in the past our knowledge of this factor has been limited by the lack of inclusion of females in appropriate studies.

Taken together, evidence from both cardiovascular and neurological research illustrates that the loss of VEGF and dopamine coordination contributes to deterioration of cardiac tissue, reduced neuronal plasticity, and heightened vulnerability to both neurodegenerative and cardiovascular diseases. Recognizing this interdependence provides a foundation for targeting shared mechanisms of decline in aging-related disorders. Given that lifestyle interventions such as regular physical activity and a healthy diet are already widely acknowledged for their capacity to promote cardiovascular and cognitive health, the next section will explore how these non-invasive strategies may naturally influence VEGF and dopamine levels, offering an additional pathway to support neurovascular resilience in aging.

## 5. Lifestyle Effects on VEGF and Dopamine

Non-pharmacological interventions are increasingly recognized as effective strategies for slowing age-related decline. Research indicates that lifestyle modifications, particularly physical activity and nutritional adjustments, can regulate VEGF and dopamine levels [41,42,43]. For example, aerobic exercise has been shown to enhance neuroplasticity and improve cognitive performance through mechanisms that increase VEGF expression and dopamine signaling. Exercise also stimulates growth factor signaling, including VEGF, while dampening neuroinflammatory pathways, factors that promote neuronal survival and synaptic plasticity [44]. Additionally, these interventions have been associated with reduced inflammatory signaling and improved functional capacity, suggesting broader systemic advantages beyond neural outcomes [45]. A meta-analysis of 18 randomized controlled trials found the most significant cognitive benefits in tasks related to executive control functions are often linked to dopaminergic regulation. In addition to cognitive enhancements, physical activity is linked with a reduced risk of chronic conditions such as hypertension and type 2 diabetes, which are known to exacerbate age-related cognitive and vascular decline [46].

Building on this evidence, exercise supports neuroplasticity and promotes vascular and neural repair during aging. Aerobic activity has been shown to increase VEGF expression in the brain, facilitating angiogenesis and supporting the maintenance of cerebrovascular integrity. In an experimental study, López-López et al. found that treadmill training significantly elevated VEGF mRNA levels in several brain regions of rats, including the hippocampus and cerebellum—areas associated with learning, memory, and motor control. The study concluded that exercise-induced VEGF signaling contributes to vascular remodeling and may help preserve brain health during aging [41]. In addition, physical activity enhances dopaminergic function, which is critical for executive processing and motor regulation. Lau et al. demonstrated that treadmill exercise increased striatal dopamine receptor expression and tyrosine hydroxylase levels in rats with Parkinsonian symptoms, improving motor performance. These results suggest that exercise may help maintain dopaminergic tone, buffering against the cognitive and motor deficits typically accompanying aging and neurodegenerative disease progression [42]. Furthermore, a 2021 review of 15 studies regarding a wide range of demographics concluded that physical activity interventions would greatly improve dopamine levels and become an innovative approach for improving mental health [47].

Nutritional interventions also influence VEGF and dopamine activity through anti-inflammatory and antioxidant mechanisms. Joseph et al. reviewed evidence that polyphenol-rich foods, such as berries, green tea, and dark leafy vegetables, can attenuate oxidative stress and modulate intracellular signaling cascades involved in neuronal resilience, synaptic plasticity, and vascular regeneration [43]. These compounds support a healthy rate of VEGF-driven angiogenesis and enhance dopaminergic transmission, reinforcing neural circuits essential for cognitive performance, mood regulation, and motor control. The combination of these effects underscores the importance of maintaining healthy lifestyle habits, as it relates to molecular systems involved in aging. By modulating VEGF and dopamine signaling, physical activity and nutrition offer a biologically grounded, low-risk approach to delaying cognitive decline, maintaining vascular health, and promoting a more resilient aging trajectory.

Although pharmacological strategies such as dopamine agonists and anti-VEGF agents remain essential in clinical settings, they are limited by systemic side effects and diminishing returns over time. For instance, dopamine agonists, while effective in symptom management for Parkinson’s disease, can lead to adverse outcomes such as excessive daytime sleepiness, hallucinations, and impulse control disorders, including compulsive gambling or hypersexuality [48]. Similarly, a pharmacovigilance study analyzing over 18 million spontaneous safety reports found a statistical signal linking intravitreal ranibizumab, an anti-VEGF therapy, to increased reports of dementia and Parkinson-like syndromes, suggesting a possible risk of cognitive and motor decline following chronic anti-VEGF exposure [49].

In contrast, as described in Figure 2, lifestyle interventions offer durable, low-risk modulation of critical aging pathways. The evidence suggests that behavioral approaches, by maintaining VEGF and dopamine function, represent foundational tools for extending health span and delaying the onset of age-related disorders. Building on the evidence that exercise, diet, and other behavioral interventions can regulate VEGF and dopamine pathways to support brain and vascular health, researchers increasingly recognize these molecules’ importance in the aging process. However, despite growing interest, the existing literature on their combined influence remains limited. This article addresses that gap by bringing greater awareness to the therapeutic potential of VEGF and dopamine signaling in aging. In this context, the next section will explore emerging therapeutic strategies and ongoing research efforts to support VEGF and dopamine signaling pathways to improve health during aging.

## 6. Therapeutic Implications

Targeting the VEGF and dopamine pathways presents both promising and complex therapeutic opportunities for managing pre-existing cardiovascular and neurodegenerative diseases. Despite their significance, few clinical interventions address their combined function, leaving a gap in therapeutic integration. Bridging this divide between experimental potential and clinical application is essential for developing more effective strategies for aging-related decline.

Dopamine-based therapies, particularly dopamine agonists such as pramipexole and ropinirole, are standard treatments for Parkinson’s disease. These agents mimic the activity of endogenous dopamine to restore dopaminergic tone in the context of progressive neuronal loss [55]. While they are effective in mitigating motor symptoms, their utility is specific to Parkinsonian pathology and not broadly applicable to age-related neurodegeneration. Though some studies suggest these agents influence intracellular pathways associated with neuronal health, their benefits beyond dopamine replacement in Parkinson’s disease are not well established [56]. Moreover, their long-term use is associated with significant side effects, underscoring the need for safer and more targeted alternatives.

In contrast, VEGF-targeted therapies have seen widespread use in fields like oncology and ophthalmology, largely for their anti-angiogenic effects. Bevacizumab, a monoclonal antibody, has effectively suppressed pathological angiogenesis in cancer models [57]. In a study involving 813 patients with untreated metastatic colorectal cancer, those who received a combination of irinotecan, bolus fluorouracil, and leucovorin (IFL) with bevacizumab experienced a median survival duration of 20.3 months compared to 15.6 months in the placebo group, an improvement of over 30% in survival outcomes. While this does not constitute a cure, it emphasizes the potential efficacy of anti-VEGF therapies in modulating angiogenesis where VEGF overexpression acts as a pathological driver [57]. However, this usage is distinct from the therapeutic goals in aging, where VEGF enhancement may be necessary to support neural repair. Evidence from disease-specific studies suggests that VEGF may contribute to neural stability through its vascular support mechanisms. However, further research is needed to confirm its neuroprotective role in aging contexts [58]. Nevertheless, translating these findings to clinical practice remains challenging due to risks like ectopic vessel growth and blood–brain barrier disruption. Emerging delivery technologies, such as gene therapies utilizing targeted nanoparticles, have shown promise in enhancing localized VEGF delivery, potentially mitigating risks associated with systemic exposure [59]. Additionally, ensuring the homeostatic balance of VEGF in the body is especially important in pharmacological VEGF therapy due to the possible inflammatory properties of VEGF. A study was conducted with the use of nine transgenic mice, aimed at inducing chronic overexpression of VEGF in basal epidermal keratinocytes via a keratin 14 promoter expression cassette. This study revealed that VEGFR-2 was significantly overexpressed and VEGFR-1 was moderately expressed, which is consistent with the behavior of many inflammatory skin diseases [60]. This could reveal that VEGF has inflammatory properties relevant to the integumentary system that could be applicable to cardiovascular and neurological pathways. Receptor-specific modulation strategies, aimed at isolating D1 (D1R and D5R) or D2 receptor activity, are under development and could allow for greater control over dopamine-related effects without inducing widespread behavioral disturbances [61]. Additionally, certain dopaminergic medications used to treat psychiatric conditions, particularly antipsychotics, have been associated with increased cardiovascular complications like cardiac valve regurgitation, arrhythmia, and fibrotic disorders [62,63]. These effects are thought to arise from their influence on the autonomic nervous system, which regulates heart rate and vascular tone. Alterations in autonomic balance may also affect endothelial signaling cascades, potentially disrupting VEGF-mediated vascular homeostasis [64].

Advances in neurovascular imaging have significantly enhanced our ability to visualize and quantify the interplay between VEGF and dopamine systems. This offers critical insights for understanding these pathways in vivo, with significant implications for aging and neurodegenerative disease. PET imaging has been successfully employed to track the delivery and biodistribution of antibodies, which allows precise monitoring of therapeutic engagement and vascular remodeling in real time [65]. Additionally, novel imaging probes targeting VEGF receptors have been developed to assess receptor expression in pathological conditions, including cancer and, potentially, age-related vascular decline, which offers a non-invasive window into VEGF signaling dynamics [66]. On the neural side, high-resolution imaging of the dopaminergic system, such as the dopamine transporter (DAT) SPECT and PET with the usage of radiotracers, is emerging in the diagnosis and progression tracking of neurodegenerative diseases like Parkinson’s, dementias, Huntington’s, and others [67]. Importantly, these imaging techniques can be leveraged together to explore the crosstalk between VEGF and dopamine systems. For example, these imaging strategies could be used to identify regions where dopaminergic decline coincides with impaired angiogenesis, which would help guide the development of co-targeting therapies.

While robust clinical data is lacking, preliminary studies suggest that dopamine agonists may regulate angiogenesis, potentially intersecting with VEGF pathways in pathological settings [28]. Significantly, these findings are based on cancer models and should be interpreted cautiously in the context of aging. Nonetheless, future therapies that jointly stabilize dopamine signaling and enhance VEGF-mediated repair may offer multifaceted protection against neurodegeneration and vascular dysfunction. A deeper understanding of VEGF–dopamine interactions will be essential for developing safe and effective dual-targeted therapies. Integrative treatments addressing aging as a multisystemic process hold the most significant promise for extending quality of life in later years and mitigating the effects of accelerated aging, particularly by preserving neurovascular function, reducing inflammation, and maintaining cognitive and cardiovascular resilience. Building on this broader physiological framework, the following section explores future research directions that may illuminate new pathways for intervention and highlight critical gaps in existing knowledge regarding VEGF and dopamine co-regulation in aging.

## 7. Perspectives

Current scientific discourse surrounding VEGF and dopamine in aging-related degeneration reveals a spectrum of interpretations, each contributing unique insights to a highly complex biological interplay. While the previous sections have examined their roles and interactions, this section aims to provide additional nuance by contextualizing these molecules within broader research frameworks and disciplinary priorities. Understanding how different fields approach these pathways reveals both promising directions and cautionary insights that can guide future investigations.

Before unified VEGF and dopamine approaches can be established, several paths must be investigated to see the effects the combination could have on older adults. One major inquiry center is defining the threshold levels of VEGF and dopamine signaling that support tissue regeneration without triggering maladaptive angiogenesis or neurotoxicity [68,69]. Moreover, researchers must investigate how intrinsic factors such as genetic predisposition, metabolic state, and age-related comorbidities affect the VEGF and dopamine interaction in individuals. There is also a growing need to evaluate the long-term efficacy and safety of various delivery methods, including systemic, localized, and receptor-specific platforms, particularly in aged tissues altered by pharmaceutical interventions. Finally, determining whether these interventions can be personalized based on individual VEGF and dopamine signaling activity and risk factors will be critical for developing safe and effective therapies.

Based on the current literature, seeking a novel therapeutic approach to maintaining equilibrium in relation to VEGF in aging persons, while ensuring that pathological conditions are not exasperated, continues to be a challenge. Figure 2 offers a conceptual framework that integrates these perspectives into a comprehensive classification of current and emerging therapeutic strategies. It illustrates how lifestyle [13,41,44,45,46], pharmacological [14,50,51], and experimental interventions [52,53,54] reflect the diverse approaches that could be used to modulate VEGF and dopamine signaling to improve neurovascular aging [13,14,54].

In summary, these diverse perspectives converge on a central insight: effective therapeutic strategies for aging must integrate the reciprocal regulation of VEGF and dopamine. Rather than treating them as independent targets, future research should prioritize their co-regulation within physiological and pathological aging processes. Recognizing the regenerative capacity and the intricate regulatory roles of VEGF and dopamine can inform more personalized and responsive therapeutic strategies. Such approaches may better address the multifaceted nature of the neurovascular decline observed in aging populations. Building on these interdisciplinary insights, the following conclusion will synthesize the key findings of this paper and propose future directions for advancing the study of VEGF and dopamine in aging physiology.

## 8. Conclusions

Aging is one of the most pressing biomedical challenges of the 21st century, a multifaceted biological process that affects every organ system and contributes to widespread vulnerability to chronic disease. Understanding the factors that influence the pace and severity of aging is essential for improving health outcomes across the lifespan. Some of the most compelling contributors to this process are vascular endothelial growth factor (VEGF) and dopamine, key regulators of vascular integrity and neural function. Although each supports essential physiological functions independently and they are not the only signaling molecules in the human body that influence aging, emerging evidence suggests that VEGF and dopamine’s combined signaling may have a significant influence on the trajectory of age-related decline. Despite this potential, the intersection of these two pathways remains significantly underexplored.

This underrepresentation in current research presents a meaningful opportunity to uncover novel targets for intervention, particularly in the context of pathological aging. As the field advances toward more preventive and personalized care strategies, interventions considering the co-regulatory influence of VEGF and dopamine may prove especially effective. Lifestyle-based approaches such as exercise and diet are already known to positively affect these pathways. At the same time, newer strategies, including receptor-specific pharmacology and gene-targeted therapies, offer the potential for highly tailored treatment. Focusing future research on the integrated role of VEGF and dopamine can help discover how their interaction contributes to age-associated vascular and neurological decline. This growing body of knowledge could also aid in identifying key biological targets and informing the development of strategies to slow the progression of accelerated aging. In doing so, the field may move closer to developing targeted interventions delaying the onset of degenerative conditions and support life-long sustained health and function.

## Figures and Tables

**Figure 1 cells-14-01178-f001:**
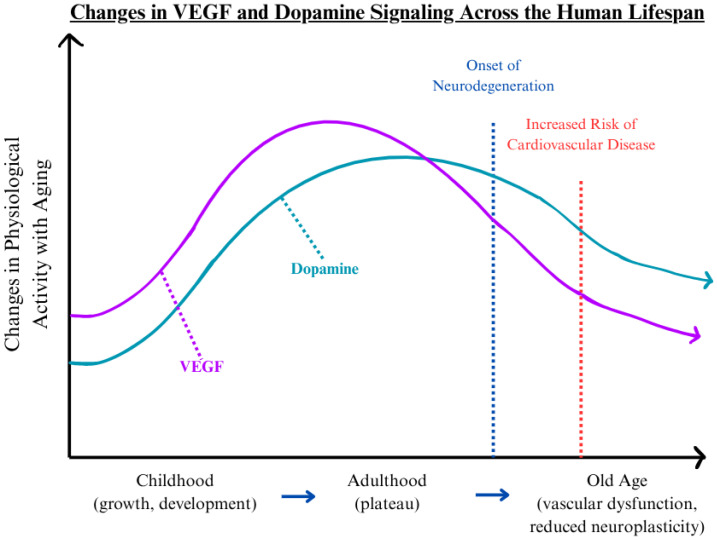
Changes in VEGF and dopamine biological activity across the human lifespan. VEGF (purple line) and dopamine (teal line) demonstrate the changes in activity of the two molecules across the human lifespan (childhood [16,17], adulthood [18,19], and old age [20,21]). The blue vertical line indicates the typical onset of neurodegenerative conditions [22,23], while the red dashed line represents the threshold of an increased risk of cardiovascular disease [24,25]. These overlapping downturns emphasize the coordinated roles of these pathways in neurovascular aging and identify a critical stage for intervention.

**Figure 2 cells-14-01178-f002:**
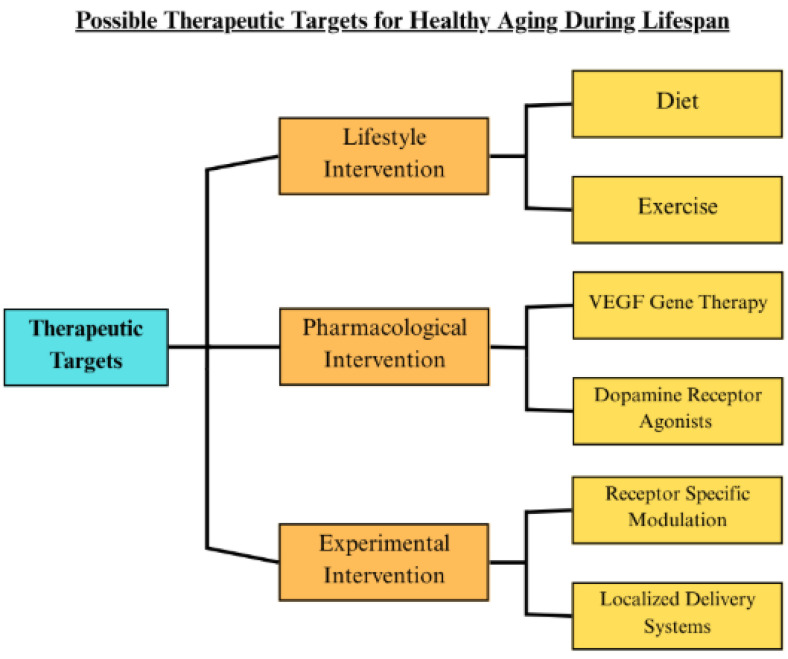
Potential therapeutic targets for healthy aging individuals. A schematic representation of therapeutic targets, categorized by lifestyle ([41,44,45,46]), pharmacological (VEGF gene therapy [14], dopamine receptor agonists [50,51]), and experimental interventions (receptor-specific modulation [52,53], localized delivery systems [54]) aimed at potentially modulating VEGF and dopamine signaling in late adulthood to improve healthy neuro- and cardiovascular aging.

## Data Availability

No new data were created.

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
