# Peer review of "Essential Pieces of the Puzzle: The Roles of VEGF and Dopamine in Aging"

_cells, 2025, doi:10.3390/cells14151178_

Round 1
Reviewer 1 Report
Comments and Suggestions for Authors
Aging is a complex process marked by declining function and higher disease risk, especially in the cardiovascular and nervous systems. Key molecular players like VEGF and dopamine play vital roles in maintaining neurovascular health—VEGF supports angiogenesis and repair, while dopamine regulates vascular tone and may influence VEGF activity. Although their individual roles in aging and disease are well-studied, their interaction, particularly in healthy versus unhealthy aging, remains poorly understood. This review emphasizes the importance of exploring VEGF-dopamine crosstalk and suggests that lifestyle interventions could modulate these pathways, offering potential strategies for promoting healthy aging and treating age-related conditions.
The review was well written, with only one minor suggestion. The subtitle "VEGF and Dopamine in Neurodegenerative Diseases" may be too broad, as the discussion focuses specifically on Parkinson’s and Alzheimer’s diseases. Therefore, I suggest revising the subtitle to "VEGF and Dopamine in Parkinson’s and Alzheimer’s Diseases" for greater accuracy.
Author Response
Reviewer 1
Comment 1: Aging is a complex process marked by declining function and higher disease risk, especially in the cardiovascular and nervous systems. Key molecular players like VEGF and dopamine play vital roles in maintaining neurovascular health—VEGF supports angiogenesis and repair, while dopamine regulates vascular tone and may influence VEGF activity. Although their individual roles in aging and disease are well-studied, their interaction, particularly in healthy versus unhealthy aging, remains poorly understood. This review emphasizes the importance of exploring VEGF-dopamine crosstalk and suggests that lifestyle interventions could modulate these pathways, offering potential strategies for promoting healthy aging and treating age-related conditions.
The review was well written, with only one minor suggestion. The subtitle "VEGF and Dopamine in Neurodegenerative Diseases" may be too broad, as the discussion focuses specifically on Parkinson’s and Alzheimer’s diseases. Therefore, I suggest revising the subtitle to "VEGF and Dopamine in Parkinson’s and Alzheimer’s Diseases" for greater accuracy.
Answer: Thank you for pointing out the importance of this review article. We appreciate the reviewer’s thoughtful suggestions of changing the subtitle “VEGF and Dopamine in Neurodegenerative Diseases” to “VEGF and Dopamine in Parkinson’s and Alzheimer’s Diseases”. We believe that this is an appropriate change, as our subtitle of Neurodegenerative Diseases is too general. We have made this change in line 217.
Reviewer 2 Report
Comments and Suggestions for Authors
This is a comprehensive review of VEGF, dopamine and ageing that has not yet been summarised elsewhere. Nevertheless, the authors could briefly discuss possible gender-specific differences in VEGF and dopamine signalling. While some evidence suggests that these signalling pathways are regulated by hormones and that there are sex differences in neurovascular ageing, there are few data directly linking sex, VEGF and dopamine interactions in ageing. Highlighting this gap would strengthen the manuscript further and provide an important direction for future research.
Author Response
Reviewer 2
Comment 1: This is a comprehensive review of VEGF, dopamine and ageing that has not yet been summarised elsewhere. Nevertheless, the authors could briefly discuss possible gender-specific differences in VEGF and dopamine signalling. While some evidence suggests that these signalling pathways are regulated by hormones and that there are sex differences in neurovascular ageing, there are few data directly linking sex, VEGF and dopamine interactions in ageing. Highlighting this gap would strengthen the manuscript further and provide an important direction for future research.
Answer: We appreciate the reviewer’s insight into sex differences. We agree that sex is an important factor in modulating the signaling pathways of both VEGF and dopamine, and that this would affect both diagnosis and treatment of neurodegenerative and cardiovascular diseases between sexes. Accordingly, we have made these changes in lines 266-302 and added references #62, 63, 64, 65, 66.
Reviewer 3 Report
Comments and Suggestions for Authors
The authors address an important and timely topic with potential clinical implications. While the evidence base for direct VEGF-dopamine interactions is limited, the authors make a reasonable case for this being an important area for future investigation. The review would benefit from more critical analysis of current evidence limitations and clearer mechanistic explanations, but overall provides a valuable synthesis that could stimulate important future research. What is missing is how imaging could be employed to better extract information about interaction, especially in patients. For example, delivery of anti-VEGF antibodies to the brain can be now enhanced and imaged by osmotic blood brain barrier opening (PMID: 30315146). At the same time, there are a lot of history and options for dopaminergic signaling in the brain, which could also be discussed (PMID: 6604227, PMID: 9607763). Therefore, a section on imaging could improve engagement of readership.
Author Response
Comment 1: The authors address an important and timely topic with potential clinical implications. While the evidence base for direct VEGF-dopamine interactions is limited, the authors make a reasonable case for this being an important area for future investigation. The review would benefit from more critical analysis of current evidence limitations and clearer mechanistic explanations, but overall provides a valuable synthesis that could stimulate important future research. What is missing is how imaging could be employed to better extract information about interaction, especially in patients. For example, delivery of anti-VEGF antibodies to the brain can be now enhanced and imaged by osmotic blood brain barrier opening (PMID: 30315146). At the same time, there are a lot of history and options for dopaminergic signaling in the brain, which could also be discussed (PMID: 6604227, PMID: 9607763). Therefore, a section on imaging could improve engagement of readership.
Answer: Thank you for pointing out the importance of this review article according to the limitations of the research in this important aspect of aging. We appreciate the reviewer’s discernment and bringing to our attention that novel imaging techniques are a critical aspect of diagnosis and treatment, especially in aging. With new noninvasive technology, imaging can certainly be used to better assess the roles of these signaling pathways as well as aid with diagnosis. We have made these changes in lines 431-446 and added references # 67, 68, 69.
Round 2
Reviewer 3 Report
Comments and Suggestions for Authors
The authors addressed my comments.